# The association between depressive symptoms and self-reported sleep difficulties among college students: Truth or reporting bias?

Zhiyong Huang[1], Fabrice Kämpfen[2]*

**1** Southwestern University of Finance and Economics, Chengdu, China, **2** Population Studies Center, University of Pennsylvania, Philadelphia, Pennsylvania, United States of America

☯ These authors contributed equally to this work.

* kampfenf@sas.upenn.edu

**Data Availability Statement:** All relevant data are within the manuscript and its Supporting information files.

## Abstract

The strong association between self-reported sleep difficulties and depressive symptoms is well documented. However, individuals who suffer from depressive symptoms could potentially interpret the values attached to a subjective scale differently from others, making comparisons of sleep difficulties across individuals with different depressive symptoms problematic. The objective of this study is to determine the existence and magnitude of reporting heterogeneity in subjective assessment of sleep difficulties by those who have depressive symptoms. We implement an online survey using Visual Analogue Scales and anchoring vignettes to study the comparability of subjective assessments of sleep difficulties among college students in Switzerland ($N = 1,813$). Using multivariate linear regressions and double-index models, our analysis shows that reporting heterogeneity plays only a marginal role in moderating the association between sleep difficulties and depression, irrespective of the severity of the depressive symptoms of the individuals. This suggests that unadjusted comparisons of self-reported sleep difficulties between college students are meaningful, even among individuals with depressive symptoms.

## Introduction

Sleep is associated with a wide range of mental health outcomes. Persons with higher sleep quality have greater well-being and better psychological functioning [1–9]. Insomnia and other sleep difficulties, on the other hand, are strong predictors of depression, anxiety, stress, and neuroticism, to name a few [1].

The study of the relationship between sleep difficulty and mental health is of particular relevance among college students as they transition into adulthood. College students, and more generally young adults (18-25 years of age), have been shown to be particularly prone to depression because this is a period in the life course that coincides with many important challenging transitions to adulthood such as separation from family, financial independence and

**Funding:** This study was funded by the HEC Research Fund 2017 - University of Lausanne, Switzerland (https://www.unil.ch/hec/en/home.html) awarded to FK and ZH. This study was also supported by the National Natural Science Foundation of China under the project "Quality of Life Research: Based on Capacity Approach: A Population Study in Southwestern China" (Project number 71804151) awarded to ZH. The funders had no role in study design, data collection and analysis, decision to publish, or preparation of the manuscript.

**Competing interests:** The authors have declared that no competing interests exist.

increased work-load [10–13]. At the same time, young adulthood is a period in which there are great shifts in circadian rhythm and sleep patterns [14, 15]. Problems with sleep length among college students have been associated with poor mental health such as depression and anxiety [16–21], suicidal thoughts and self-harm behaviors [22]. Given the increasing prevalence of depression among young adults in the recent years [23] and the fact that depression in young adults constitutes an important risk factor for depression and bipolar disorder during adulthood [24, 25], having a clear understanding of the association between sleep difficulties and depression is crucial if one wants to formulate lifestyle guidance and design policies to promote healthy development of college students and young adults.

Subjective sleep quality measures have been a central feature to assess insomnia, non-restorative sleep and sleep health in general [26]. Several tools and instruments have been implemented to assess the subjective sleep quality and corresponding sleep difficulties of individuals. Questionnaires such as the Pittsburgh Sleep Quality Index (PSQI), the Insomnia Severity Index, the Athens Insomnia Scale, Epworth Sleepiness Scale (ESS), the Leeds Sleep Evaluation Questionnaire (LSEQ), to name a few, are commonly used as effective means to aid diagnosis of sleep difficulty [27–31]. Moreover, single-item questions are commonly employed in large-scale population health surveys such as the Health and Retirement Study (HRS) in the USA [32], the Survey of Health, Aging and Retirement in Europe (SHARE) [33], the WHO Study on Global Aging and Adult Health (SAGE) [34], the China Health and Retirement Longitudinal Study (CHARLS) [35] to assess individual's sleep difficulties in non-clinical settings.

One particular characteristics of all these instruments is that they are based on self-reports and may therefore suffer from subjective bias. It has been shown that there exist significant differences between objective measures, such as polysomnography and actigraphy, and subjective measures of sleep difficulties [26, 36–40]. Buysse et al. (2008) [36] have attributed the weak association between subjective and objective measures of sleep to whether one assesses habitual patterns of sleep versus sleep on a discrete occasion. Baker et al. (1999) [39] have argued that the weak correlation between objective and subjective measures are due to between-individual variation while Jean-Louis and Kripke (2000) [40] have speculated that the subjective measures may be biased by the depressive mood of respondents.

As documented, inconsistencies between subjective and objective measures of sleep difficulties can arise because of several reasons, one of them being systematic between-individual reporting heterogeneity. Reporting heterogeneity, also known as differential item functioning (DIF) in the literature, arises when individuals with different characteristics interpret the possible values attached to a scale differently from one another, i.e., when they adopt different reporting scales to rate their own characteristics. For instance, one of the items of the PSQI, which is the most widely used sleep health assessment tool in both clinical and non-clinical settings [41], is phrased as "During the past month, how would you rate your sleep quality overall?", with possible answer being "very good", "fairly good", "fairly bad" and "very bad". In the case of individuals with depression, the subject of our study, one critical concern would be that respondents with depression might have very different interpretation of the words "good", "bad", "fairly" and "very". Individuals with depressive symptoms might therefore report poor sleep quality, either because they are indeed deprived of sleep as a result of depression, or simply because they are inclined to rate everything more negatively, making subjective sleep quality assessments between individuals with and without depressive symptoms hard to compare [42].

As another example, LSEQ, which is very commonly used to assess sleep difficulties [43], contains a Visual Analogue Scale (VAS) to measure sleep difficulties. VAS is a measurement instrument that is used to rate a given characteristic over a continuum of values (a horizontal or vertical line) with "anchors" at the two end-points, representing the best and worst

scenarios. The anchors in LSEQ are "more difficult than usual" as one endpoint and "easier than usual" as another. Both of these endpoints can be interpreted very differently by individuals with different depression level. And because respondents use these anchors to evaluate their sleep difficulty, the variations in the interpretations of these anchors across groups of respondents could potentially mean that they use different reporting scales, making comparisons between self-reported measures of sleep difficulty across these groups hard to interpret. Other factors than depression, such as sex and income, can also bias self-reported sleep difficulty if respondents interpret the scale of the sleep difficulty measure differently from others [44].

Because it is important to understand whether the association between a subjective measure such as sleep difficulty, and its important risk predictors –depressive symptoms in our case– still holds after controlling for reporting differences among individuals, researchers have designed techniques to account for reporting heterogeneity in self-assessed measures. One example of such techniques is anchoring vignettes [45–47], which are short descriptions of hypothetical individuals that are assessed by respondents alongside their self-assessment of the same domain. For example, an anchoring vignette on sleep patterns included in SAGE is profiled as: "[Mark] falls asleep every night within five minutes of going to bed. He sleeps soundly during the whole night and wakes up in the morning feeling well-rested." Respondents are asked to evaluate the sleep quality of this hypothetical person alongside their own sleep quality.

As anchoring vignettes are predefined and invariant across individuals, their ratings are supposed to only reflect differences in reporting scales, which in turn allows to identify DIF among respondents. Anchoring vignettes can therefore be used to adjust subjective measures by purging reporting heterogeneity from the self-assessed measures, making the adjusted self-assessment better comparable across individuals. Anchoring vignettes have been used in many research areas such as health [47–54], healthcare [55, 56], subjective well-being and quality of life [46, 57–60], political efficacy [45, 61], job satisfaction [62], working disability [63] and social status [64].

The objective of this study is to determine the existence of reporting heterogeneity in subjective assessment of sleep difficulties among college students, and explore whether this reporting heterogeneity is explained by differences in depressive symptoms as measured by the standard and validated PHQ-9 instrument. We use data collected from a sample of college students in the region of Lausanne in Switzerland whom we asked to complete an online survey. Online surveys are particularly well-adapted for this kind of research question [63] and have been used in many other studies to assess reporting heterogeneity in self-reported evaluations [63, 65, 66]. We hypothesize that college students with and without depressive symptoms adopt different reporting scales when assessing their sleep difficulties, to an extent that is hard to evaluate a priori. To test our hypotheses, we first document the unadjusted associations between depressive symptoms and self-assessed sleep difficulties. We then exploit five sleep-related anchoring vignettes that are derived from the existing literature to adjust for reporting heterogeneity in the relationship between self-assessed sleep difficulties and depressive symptoms. The econometric specification we put in place will allow to (i) disentangle the "true" difference in sleep difficulties from reporting heterogeneity and (ii) uncover the individual characteristics that are driving this reporting heterogeneity. In addition to various socio-demographic information that we collected as part of our survey, our econometric models will also control for various measures of sleep patterns that could confound the relationship between sleep difficulties and depressive symptoms, as explained below.

Although some studies have examined the association between depressive symptoms and self-assessed sleep quality and difficulties [2, 67, 68], very few have attempted to assess the existence of reporting heterogeneity in subjective sleep assessment and to correct for it. Tareque

et al. (2016) [44] seems to be the only exception. However, Tareque et al. (2016) [44] do not look specifically at depression as the main source of reporting heterogeneity and they measure sleep difficulties on a Likert scale, making interpretation of the magnitude of reporting heterogeneity difficult. Using self-assessed sleep difficulties and anchoring vignettes measured on VAS, we can evaluate the magnitude, if any, of reporting heterogeneity in self-assessed sleep difficulties. In addition, our study is the first to investigate the existence of reporting heterogeneity in subjective assessment of sleep difficulties among college students, which, as discussed above, is a strata of the population that is particularly prone to both sleep difficulties and mental disorders.

## Methods

### Participants

We recruited our sample of college students from three schools of higher education in Canton de Vaud in Switzerland (the Swiss Federal Institute of Technology in Lausanne (EPFL), the University of Lausanne (UNIL) and Ecole hoteliere de Lausanne (EHL)). College students from these three schools can register to the Laboratory of Behavioral Experiments program at the University of Lausanne and participate in experiments and surveys. The online questionnaire we used was created on Qualtrics and sent out to all students who registered in the University of Lausanne's experiment program. The invitation email we sent to students and the introductory page of our online survey did not mention the specific aim of the study but referred to it as a study on their quality of life in general. 6,578 emails were sent, out of which 1,938 students completed our survey in the following two weeks (response rate = 29.5%). For their participation, students who completed our online survey were automatically entered into a lottery for which the highest prize was about USD 300. The experimental protocol and informed consent were approved by HEC Ethics Committee of the University of Lausanne in February 2017, and all subjects gave informed consent.

Out of the 1,938 college students who completed our online questionnaire, we kept those who were between 18 and 25 years old to capture "standard" college progression. This age restriction also corresponds to late adolescent/young adulthood [69], which represents a period in life with important transitions from adolescence to adulthood [70] that coincide with major changes both in mental health and sleep patterns/circadian rhythm [10–15]. Out of 1,938 students who answered our survey, only 3 respondents were 17 years old (as compared to 160 college students who were 18 years old), while 78 respondents were 25 years old, as opposed to 39 respondents who were 26. After discarding 125 observations that were out of the age range, our final sample consisted of 1,813 observations.

### Survey design and measurements

After completing the consent forms, college students were first asked to evaluate our set of five sleep-related anchoring vignettes using a VAS ranging from 0 ('no difficulty') to 100 ('extreme difficulty'). We then asked them to evaluate their own sleep difficulties, using the same VAS. As discussed in Hopkins and King (2010) [61], asking individuals to evaluate anchoring vignette first allows to prime respondents into interpreting the self-assessment question in a similar light and defining the response scale in a common way. The self-assessment question was then followed by a series of more general questions on socio-demographic characteristics from which we derive information to create our set of control variables.

**Measures on subjective sleep difficulties and anchoring vignettes.** Our subjective measure of sleep difficulties was derived from the respondents' answer to the question "Overall in the last 30 days, how much of a problem did you have with sleeping, such as falling asleep,

waking up frequently during the night or waking up too early in the morning?", which is directly taken from the SAGE/World Health Survey designed by the World Health Organization (WHO) [71] and very similar in essence to Tareque et al. (2016) [44] to facilitate comparisons. Respondents had to use a VAS that was operationalized online as a horizontal graphic slider on which they could select any point among the continuum set of possible values ranging from 0 ('no difficulty') to 100 ('extreme difficulty'). These two anchors are the corresponding labels to the Likert scale used by the WHO in the World Health Survey ('none' and 'extreme') [71]. Only the minimum and maximum values at both ends of the horizontal graphic slider, 0 and 100, were displayed and no tick points in between were shown.

The five sleep-related anchoring vignettes we use in our study are taken from the World Health Survey [71] developed by the WHO as well. Widely used and validated, the original WHO vignettes were intended for adult populations and not for college students. Hence, to fit our college student population, we made a trivial change to one of the vignettes in which the expression "s/he is late to *work* 4 days out of 5" in the original text was modified to "s/he is late to *school* 4 days out of 5". The vignettes that were developed by WHO and that we use in our study include several factors associated with sleep difficulties such as time it takes for individuals to fall asleep, the quality of the sleep, the difficulty the person has to wake up as well as whether the person feels well-rested in the morning. Moreover, the vignettes respondents have to evaluate are sex-specific, matching the sex of the respondent to the sex of the fictitious persons described in the hypothetical scenarios. As documented by Jürges and Winter (2013) [72], the sex of the persons described in the vignettes could potentially be a confounder when vignettes are evaluated. Therefore, survey including vignettes usually either assign gender-specific vignettes to respondents [45, 49, 73, 74] or control for sex in all estimations [63, 65]. We follow the literature and design sex-specific vignettes and control for sex in our analysis. The five vignettes we use in our study can be found in S1 Appendix.

In addition to self-assessed sleep difficulties and the five vignette evaluations, college students were further asked questions about their sleep patterns, such as the number of effective hours of sleep they had per night on average over the past 30 days, at what time they usually went to bed, how long it took them to fall asleep and what time they woke up in the morning. All these factors are strongly correlated with both depression and sleep difficulties [75] and can therefore confound the association between these two variables. We will control for these factors in our econometric models and show that adding these sleep patterns in our specifications do not alter our results.

**Measures on depressive symptoms.**   We assess the presence and severity of depressive symptoms in our sample of college students using the standard Patient Health Questionnaire-9 (PHQ-9). The PHQ-9 is a validated 9-item instrument that is frequently used to screen for depression in primary care and non-clinical settings [76–78], including the screening of depression among college students [79, 80]. It is worth noting that its computerized version has also been validated [81]. The PHQ-9 can be found in S2 Appendix.

Its score ranges from zero to 27. Following Kroenke et al. (2001) [82], we categorize our respondents as having "no or minimum symptoms" if their score is between 0 and 9, "mild depressive symptoms" if between 10 and 14, "moderate depressive symptoms" if between 15 and 19 and "severe depressive symptoms" if above 19 points. The Cronbach value of the PHQ-9 in our sample equals to 0.791, indicating good reliability of the items to measure the same construct [83]. As a robustness check, we present results when PHQ-9 score instead of depressive symptoms is used as a measure of mental health. We show that our results when analyzing the presence of depressive symptoms are similar when analyzing depressive symptom severity.

**Measures on socio-demographic characteristics.**   We obtained information on our respondents about their age, nationality, relationship status (in couple or single), quality of the

relationship with their parents as well as their parents' education level, family income, school performance, and the number of close friends. These are factors that can explain both sleep difficulties and depression [20, 84] and we hence include these characteristics as control variables in our econometric specification.

## Statistical methods

To gain a preliminary understanding of the existence and strength of the correlation between sleep difficulties and depressive symptoms in our college student population, we first use Epanechnikov kernel densities to investigate the bivariate relationship between these two variables. We further investigate the relationship between sleep difficulties and depressive symptoms using multivariate linear regressions, including various socio-demographic characteristics and self-reported measures of sleep patterns that are not subject to reporting heterogeneity, e.g., hours of sleep, as control variables. The use of linear specifications is supported by various tests we implement and report in S5 Appendix.

We then explore two different ways to determine the existence and influence, if any, of reporting heterogeneity in subjective sleep difficulties. Perhaps the first and most straightforward way to control for reporting heterogeneity in our setting is to include vignette evaluations as additional controls in the linear regressions. This assumes that respondents use the same reporting scale when evaluating their own sleep difficulties as the scale they use when evaluating the sleep difficulties of the person described in the anchoring vignettes. This strategy, however, does not allow us to determine what the characteristics of the individuals that drive reporting heterogeneity are. Another way to disentangle reporting heterogeneity and sleep difficulties is to perform a double-index model [60]. This model exploits respondents' evaluations of sleep difficulties for themselves and for vignettes, assuming that individuals use the same reporting scales. Under this assumption, one can control for reporting heterogeneity as one controls for individual fixed effects in panel data models (where several observations are recorded for the same individual over time). The advantage is that the model allows us to not only control for reporting heterogeneity but also estimate the characteristics of the individuals that drive reporting heterogeneity.

More formally, we assume $y_{i0}^*$, the latent level of sleep difficulties of respondent $i$, can be modeled as a linear function of a set of factors $x_i'$ subject to an error term $\epsilon_{i0}$:

$$y_{i0}^* = x_i'b_0 + \epsilon_{i0} \tag{1}$$

When the level of sleep difficulties $y_{i0}^*$ is self-reported, the measurement can be potentially biased due to reporting heterogeneity, i.e., respondents with different characteristics report value of $y_{i0}^*$ in systematically different ways. By assuming that reporting heterogeneity can take the form of an additive and unobserved individual effect $c_i$, the reported value of sleep difficulties of respondent $i$, $y_{i0}$, takes the form:

$$y_{i0} = y_{i0}^* + c_i \tag{2}$$

In addition to these assumptions, the use of anchoring vignettes to correct for reporting heterogeneity requires the very common vignette-related assumptions, namely, vignette equivalence and response consistency [45, 47, 55, 62, 63]. Vignette equivalence assumes that vignettes are perceived in the same way by all respondents up to an idiosyncratic error term. Under this assumption, the perception of vignette $j$ is:

$$y_{ij}^* = b_j + \epsilon_{ij} \quad \text{for} \quad j \geqslant 1 \tag{3}$$

where $b_j$ is the location of vignette $j$. On the other hand, vignette consistency assumes that

respondents' evaluations of the anchoring vignettes are subject to the same reporting heterogeneity as their self-reported variable of interest (see Eq 2), i.e.,:

$$y_{ij} = y_{ij}^* + c_i \quad \text{for} \quad j \geqslant 1 \tag{4}$$

By defining $c_i$ as a linear function of $x_i$, i.e., $c_i = x_i'\gamma$, we can plug in $y_{i0}^*$ into Eq 2 and $y_{ij}^*$ into Eq 4 to obtain the following system of equations:

$$y_{i0} \quad = x_i'b_0 + x_i'\gamma + \epsilon_{i0} \tag{5}$$

$$y_{ij} \quad = b_j + x_i'\gamma + \epsilon_{ij} \quad \text{for} \quad j \geqslant 1 \tag{6}$$

We can then simultaneously estimate this system of equations with OLS, where $b_0$ represents the "true" effect of $x$ on subjective sleep difficulties and $\gamma$ represents the effects of $x$ on reporting heterogeneity $c$.

## Results

### Descriptive statistics

Table 1 reports the descriptive statistics of our study sample. The average score of sleep difficulty is 37.7, on a scale where 0 represents "no difficulty" and 100 "extreme difficulty". Up to 16%, 6% and 1.8% of the college students in our sample have mild, moderate, or severe depressive symptoms, respectively, as reflected by their PHQ-9 score. Females account for 51% of our sample. More than half (52%) of our respondents are Swiss, and a significant portion of them are coming from Western Europe (35%) and Africa (7.3%). Most of the respondents are undergrad students (82%). 34% of our respondents are from families that are making between USD 7,000 and 12,000 per month, which is comparable to the average gross income per month per household in Switzerland in 2014 (about USD 10,0798). 51% of our respondents claim they are single at the time of the online survey, 2.4% admit to have no close friends and 7.7% have no siblings. Among our respondents, about 3% of them evaluate the relationship with their parents as being bad. More than half of them consider their school performance as average and another 13% think their performance to be below average.

Fig 1 shows the Epanechnikov kernel density of the self-assessed sleep difficulty of the college students in our sample. This distribution appears to be highly bimodal, with the middle inflection point at about 40. The majority of respondents in our sample appear to have mild sleep difficulties, where the density distribution is the highest at roughly 15. There is also a large share of our respondents who appear to have difficulties with their sleep, as evidenced by the large number of evaluations that are above 50.

Fig 2 reports the distributions of the five anchoring vignette evaluations. We can see that our five vignettes cover the range of possible sleep difficulty values very well. The first plot in the first row represents the evaluation by our respondents of the first vignette. There appears to be unanimity across our respondents: the sleep pattern of the fictitious person described in vignette 1 represents an ideal scenario (most of the density is located at 0). At the complete opposite, the middle plot in the second row represents the evaluation by our respondents of vignette 5. Although somewhat more skewed towards the left, most of our sample evaluated scenario 5 as representing the scenario of someone experiencing the most severe difficulty with one's sleep pattern. Vignettes 2, 3, and 4 represent mid-range scenarios. It is interesting to note here that the evaluations of vignette 2 appear to be quite discordant, as revealed by the density which, in addition to displaying bimodality, is relatively high from value 20 to 70.

**Table 1. Descriptive statistics (*N* = 1, 813).**

| Variable | Mean | SD | Min | Max |
|---|---|---|---|---|
| Sleep difficulty (VAS score) | 37.7 | 27.4 | 0 | 100 |
| Depressive symptoms (based on PHQ-9 score) | | | | |
| No or minimum symptoms (reference group) | 0.76 | 0.43 | 0 | 1 |
| Mild | 0.16 | 0.37 | 0 | 1 |
| Moderate | 0.060 | 0.24 | 0 | 1 |
| Severe | 0.018 | 0.13 | 0 | 1 |
| Female | 0.51 | 0.50 | 0 | 1 |
| Age (in years) | 21.0 | 1.85 | 18 | 25 |
| Place of origin | | | | |
| Swiss (reference group) | 0.52 | 0.50 | 0 | 1 |
| Western Europe | 0.35 | 0.48 | 0 | 1 |
| Eastern Europe | 0.030 | 0.17 | 0 | 1 |
| Africa | 0.073 | 0.26 | 0 | 1 |
| Others | 0.033 | 0.18 | 0 | 1 |
| Education | | | | |
| Bachelor (reference group) | 0.82 | 0.38 | 0 | 1 |
| Master | 0.16 | 0.36 | 0 | 1 |
| Others | 0.020 | 0.14 | 0 | 1 |
| Family income | | | | |
| Less than USD 4000 (reference group) | 0.081 | 0.27 | 0 | 1 |
| (4000,7000] | 0.21 | 0.41 | 0 | 1 |
| (7000,12000] | 0.34 | 0.47 | 0 | 1 |
| More than 12000 | 0.26 | 0.44 | 0 | 1 |
| Don't know | 0.10 | 0.30 | 0 | 1 |
| School performance | | | | |
| Above average (reference group) | 0.27 | 0.45 | 0 | 1 |
| Average | 0.60 | 0.49 | 0 | 1 |
| Below average | 0.13 | 0.34 | 0 | 1 |
| Single (not in a relationship) | 0.51 | 0.50 | 0 | 1 |
| Close friends | | | | |
| None (reference group) | 0.024 | 0.15 | 0 | 1 |
| One | 0.041 | 0.20 | 0 | 1 |
| Two | 0.16 | 0.37 | 0 | 1 |
| Three or more | 0.78 | 0.42 | 0 | 1 |
| Siblings | | | | |
| None (reference group) | 0.077 | 0.27 | 0 | 1 |
| One | 0.46 | 0.50 | 0 | 1 |
| Two | 0.31 | 0.46 | 0 | 1 |
| Three or more | 0.16 | 0.36 | 0 | 1 |
| Relationship with parents | | | | |
| Good (reference group) | 0.66 | 0.47 | 0 | 1 |
| Good, but not always | 0.31 | 0.46 | 0 | 1 |
| Bad | 0.031 | 0.17 | 0 | 1 |

*Note*: Unweighted sample characteristics of the students who have completed our online survey in April 2017.

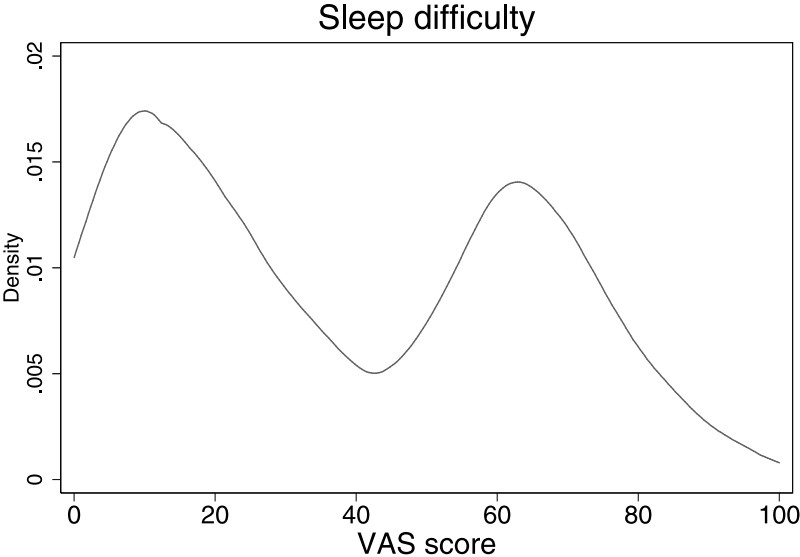

**Fig 1. Distribution of sleep difficulty evaluations.** *Note*: 0 means 'no difficulty' and 100 means 'extreme difficulty' with sleep.

## Relationship between depressive symptoms and sleep difficulties

We start our analysis by examining the bivariate relationship between self-assessed sleep difficulties and depressive symptoms. Fig 3 shows the levels of self-assessed sleep difficulties of respondents in our sample by depressive symptoms status. As can be seen from the densities, the bimodality of the distribution exists for all four depressive symptoms categories, although

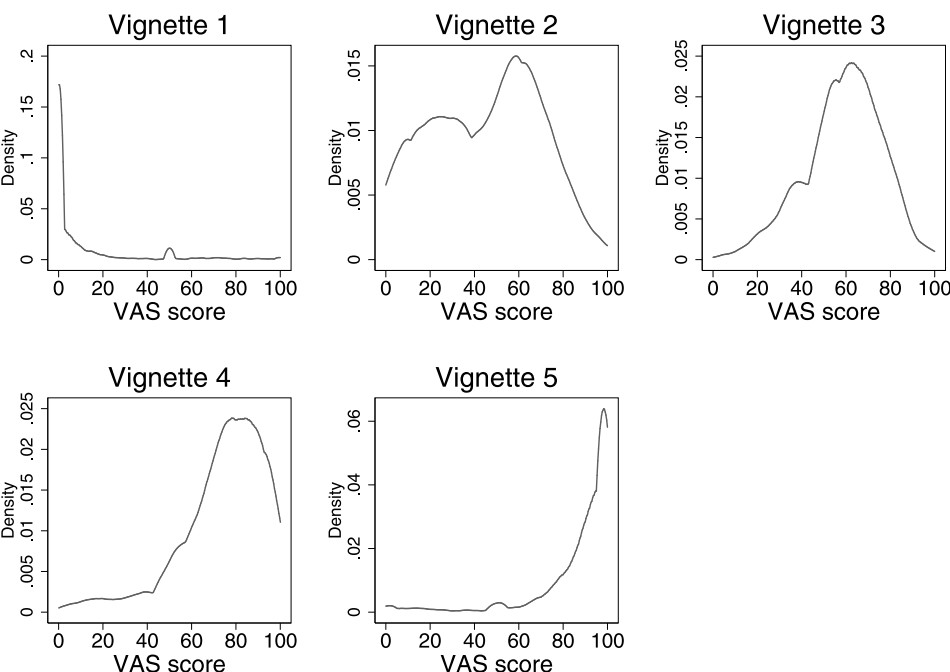

**Fig 2. Distribution of vignette evaluations.** *Note*: 0 means 'no difficulty' and 100 means 'extreme difficulty' with sleep.

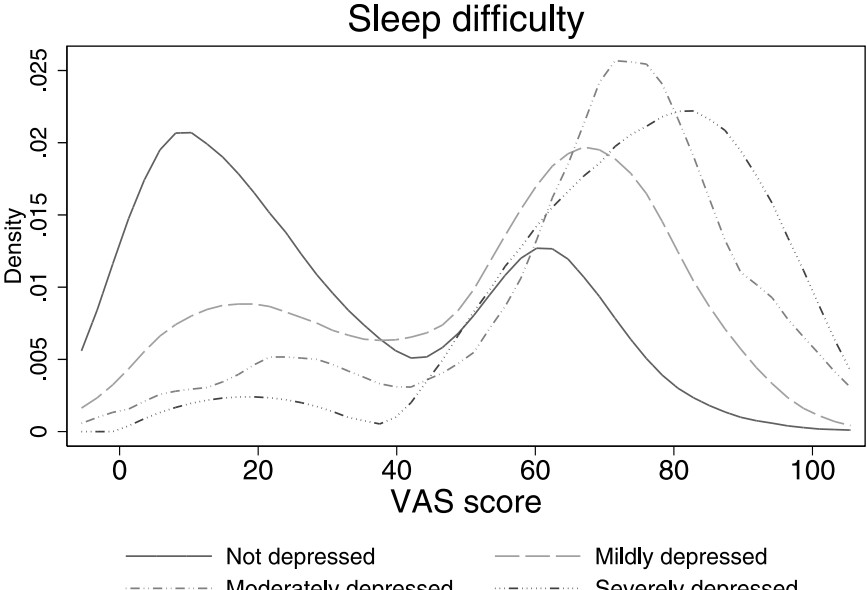

**Fig 3. Distribution of sleep difficulty evaluations by depressive symptoms.** *Note*: 0 means 'no difficulty' and 100 means 'extreme difficulty' with sleep.

it is somewhat less pronounced for those who are moderately depressed and those who are severely depressed. The relationship between sleep difficulties and depressive symptoms is very strong: almost all individuals suffering from depressive symptoms, irrespective of the severity, have difficulties with their sleep, with the density shifting towards 100 the more severe the depressive symptoms are.

Results from a multivariate linear regression of self-assessed sleep difficulties measured from 0 to 100 on our set of control variables are reported in the first column of Table 2. Sex, age, income and relationship status appear to not affect sleep difficulties. Note that we report in the tables only the coefficients associated with sex, age, and depression symptoms status, as they are the main focus of this study. Details on the coefficient of the other variables are available upon request to the authors. Individuals suffering from 'mild', 'moderate' and 'severe' depressive symptoms however have experienced great difficulties with sleep as they have rated their sleep difficulties 20, 32, and 37 points higher, respectively, compared to college students with 'no or minimal' symptoms. These effects are all significant at the level of 1% (p<0.01).

One may wonder whether the relationship between sleep difficulty and depressive symptoms could be confounded with sleep time and various patterns of sleep. Indeed, the association could be spurious if it turns out that sleep time or any other variables affecting both sleep difficulties and depressive symptoms are not controlled for in the analysis.

Columns 2, 3, 4, 5, and 6 of Table 2 however show that controlling for various measures of sleep patterns and efficiency that are commonly used in the literature does not critically affect the relationship between sleep difficulties and depressive symptoms. Descriptive statistics of these variables are presented in S1 Table. Controlling first for the reported effective hours of sleep of the respondents (column 2) [67, 75, 85, 86], which is derived from the question: *Over the last month, how many hours of sleep have you had on average every night?*", one can see that the association between depressive symptoms and sleep difficulties remains very large and significant. For instance, the increase in sleep difficulties for those suffering from severe depressive symptoms, compared to those with no depressive symptoms, only changed from 37.4

**Table 2. Linear regressions of sleep difficulties on our set of control variables.**

| | (1)<br>Sleep difficulty | (2)<br>Sleep difficulty | (3)<br>Sleep difficulty | (4)<br>Sleep difficulty | (5)<br>Sleep difficulty | (6)<br>Sleep difficulty |
|---|---|---|---|---|---|---|
| Female | 1.24 | 1.58 | 1.29 | 1.62 | 0.87 | 0.21 |
| | (1.20) | (1.18) | (1.21) | (1.20) | (1.14) | (1.09) |
| Age | 0.20 | 0.078 | 0.16 | 0.20 | 0.13 | 0.14 |
| | (0.39) | (0.38) | (0.39) | (0.39) | (0.37) | (0.36) |
| Mild depressive symptoms | 19.6*** | 18.0*** | 19.7*** | 19.2*** | 15.6*** | 14.1*** |
| | (1.69) | (1.67) | (1.69) | (1.68) | (1.66) | (1.57) |
| Moderate depressive symptoms | 32.4*** | 30.8*** | 32.5*** | 31.2*** | 21.9*** | 24.7*** |
| | (2.40) | (2.19) | (2.38) | (2.34) | (2.52) | (2.27) |
| Severe depressive symptoms | 37.4*** | 35.7*** | 37.6*** | 36.8*** | 25.9*** | 28.5*** |
| | (3.69) | (3.59) | (3.66) | (3.61) | (4.28) | (3.40) |
| Hours of sleep (reported) | | -5.26*** | | | | |
| | | (0.59) | | | | |
| Hours of sleep (wake up time—bed time) | | | -0.77 | | | |
| | | | (0.56) | | | |
| Hours of sleep, adjusted for time it takes to fall asleep | | | | -2.95*** | | |
| | | | | (0.54) | | |
| Time to fall asleep (in hours) | | | | | 25.8*** | |
| | | | | | (2.71) | |
| Take more than 20 mins to fall asleep | | | | | | 17.7*** |
| | | | | | | (1.17) |
| Sleep less than 7 hours | | | | | | 6.76*** |
| | | | | | | (1.42) |
| Low sleep efficiency | | | | | | 9.07*** |
| | | | | | | (1.48) |
| Constant | 17.5* | 59.9*** | 24.6** | 39.7*** | 9.68 | 11.9 |
| | (9.93) | (10.9) | (10.6) | (10.5) | (9.88) | (9.18) |

*Note*: Robust standard errors are in parentheses (* $p < 0.1$, ** $p < 0.05$, *** $p < 0.01$). We also control for parent's income, education level, nationality, number of siblings, relationship status, relationship with parents, number of close friends and school performance. These coefficients are not reported in the table but are available upon request.

points (p<0.01) to 35.7 (p<0.01). Controlling next for the hours of sleep defined as the difference between the time a person wakes up and the time he or she goes to bed [75], the effect of depressive symptoms again remains almost identical (column 3). In column 4, we control again for hours of sleep as defined in column 3, but this time correcting for the time it takes for respondents to fall asleep [75, 85]. Again the effect of depressive symptoms on sleep difficulties is very robust: for instance, the increase in sleep difficulties for those suffering from moderate depressive symptoms, compared to those with no depressive symptomts, only change from 32.4 points (p<0.01) to 31.2 (p<0.01), which corresponds to a change of only about 4%. The association also holds when we control only for the time it takes individuals to fall asleep instead (column 5) [67, 75, 87]. Two things are worth noting here. The first is that this variable captures some of the depressive symptoms-sleep difficulties gradients as the coefficients associated with depressive symptoms decreased by about 30%. Second, the time it takes individuals to fall asleep explains quite a lot of the sleep difficulties variation, "independently" of depressive symptoms status, given the large and statistically significant effect of the variable on sleep difficulties (about 25.8 points, p<0.01).

Finally, in column 6, we control for whether it takes more than 20 minutes for individuals to fall asleep (the usual cutoff point for sleep latency [2]) and whether college students sleep less than 7 hours per night on average [2]. In the same regression, we also add an indicator variable that reflects the sleep inefficiency of an individual as defined as taking the value of 1 if the ratio of the number of effective hours of sleep over the difference between the waking up time and the time the individual goes to bed is lower than 0.85 [75, 88–90]. Results show that including these three indicator variables in our statistical specification, although they are individually all positive and highly significant (p<0.01), do not appear to attenuate the associations between depressive symptoms and sleep difficulties.

Note that our results are robust to using PHQ-9 score as a continuous measure instead of descriptive depression categories. Indeed, as evidenced in S2 Table, the effect of a marginal increase in PHQ-9 score on sleep difficulty is equal to about 3 points (p<0.01) and remain positive and statistically significant even after including our various measures of sleep patterns and efficiency as control variables in the econometric specification.

Sleep patterns and hours of sleep, therefore, do not seem to explain the association between depressive symptoms and sleep difficulties as the association still appears to be strong and statistically significant after controlling for these different confounding variables. Could the association between depressive symptoms and sleep difficulties be explained, at least partially, by reporting heterogeneity? As explained above, individuals with depressive symptoms could report severe sleep difficulties, either because they are indeed deprived of good sleep as a result of depression or simply because they are apt to rate everything more negatively. Reporting heterogeneity could indeed stem from the fact that the meaning of the two end-points of our VAS, 'no difficulty', and 'extreme difficulty', could be interpreted very differently depending on whether a person suffers from depression or not. We now focus our analysis on reporting heterogeneity and try to answer this question.

## Reporting heterogeneity in sleep difficulties

As we are most concerned about whether individuals with depressive symptoms could report sleep difficulty on a different scale from others, we start our analysis about the presence of reporting heterogeneity by exploring the bivariate relationship between vignette evaluations (which measure reporting scale) and depressive symptoms. Fig 4 shows the distribution of the evaluations of the five anchoring vignettes used in our study by depressive symptoms severity. If the vignette equivalence assumption holds, any difference in vignette evaluations by individuals with different depressive symptoms would suggest the presence of reporting heterogeneity in sleep difficulty assessment. The five plots of Fig 4 show that most vignette evaluations are relatively homogeneous, irrespective of the depressive symptoms college students are suffering from. This holds true except perhaps among those with severe depressive symptoms, for which noticeable differences can be seen in the evaluations of vignettes 2 and 4. While purely descriptive, this bivariate analysis already shed light on the possible presence of reporting heterogeneity in self-assessed sleep difficulty among college students with and without depressive symptoms.

We then resort to multivariate analysis and regress the five vignette evaluations on our set of control variables, including depressive symptoms. Under the vignette equivalence assumption, if the coefficient of a variable turns out to be statistically significant, then it would mean that the vignettes are not perceived equally by respondents who have the characteristic associated with that particular coefficient. In other words, there would be some reporting heterogeneity associated with that specific characteristic $x$ (See Eq 6). Table 3 reports the results of the regressions of our five vignettes on our set of control variables. It is reassuring that very few

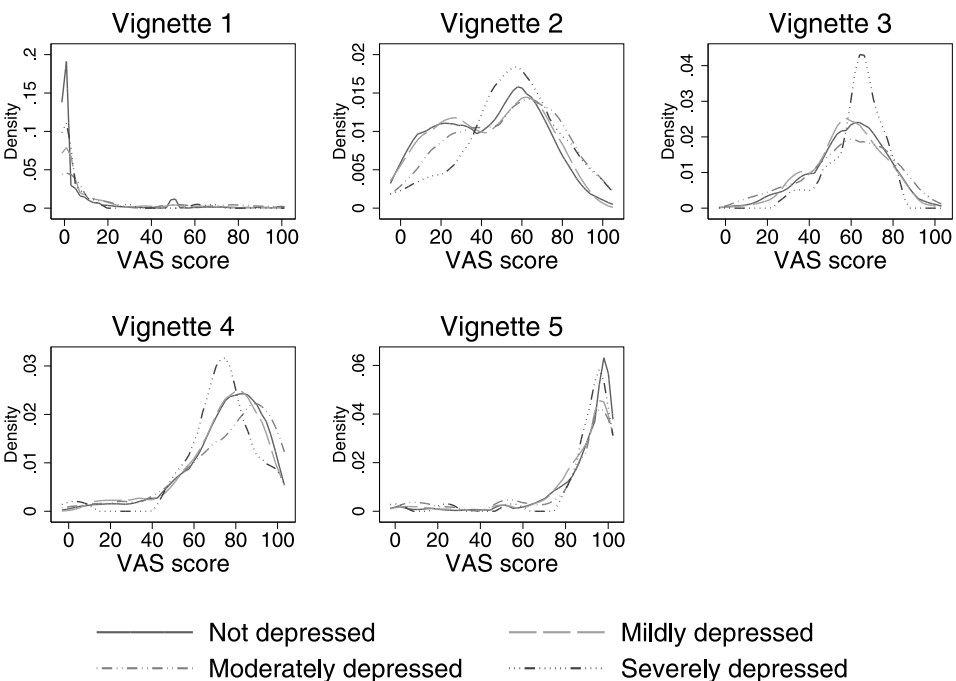

**Fig 4. Distribution of vignette evaluations by depressive symptoms.** *Note*: 0 means 'no difficulty' and 100 means 'extreme difficulty' with sleep.

coefficients are significantly different from 0, suggesting that individuals evaluate vignettes in a rather similar way. One exception clearly stands out, however. Individuals suffering from 'moderate' and 'severe' depressive symptoms appear to evaluate vignette 2 more negatively, with their evaluations being on average 8.3 ($p<0.01$) and 10.6 ($p<0.05$) points closer to 'extreme difficulty', respectively, compared to those with 'no or minimal' depressive

**Table 3. Linear regressions of vignette evaluations on our set of control variables.**

|  | (1) Vignette 1 | (2) Vignette 2 | (3) Vignette 3 | (4) Vignette 4 | (5) Vignette 5 |
|---|---|---|---|---|---|
| Female | 2.41** | 0.48 | -1.46* | -0.97 | -1.67* |
|  | (0.97) | (1.23) | (0.85) | (0.96) | (0.99) |
| Age | 0.20 | 0.33 | -0.25 | -0.26 | -0.26 |
|  | (0.31) | (0.39) | (0.28) | (0.31) | (0.31) |
| Mild depressive symptoms | 0.87 | -0.15 | -0.91 | -0.87 | -1.51 |
|  | (1.29) | (1.67) | (1.11) | (1.31) | (1.38) |
| Moderate depressive symptoms | 5.96** | 8.26*** | 0.15 | 2.09 | -3.34 |
|  | (2.78) | (2.62) | (2.17) | (2.19) | (2.56) |
| Severe depressive symptoms | -4.58 | 10.6** | 3.76* | -1.32 | 1.92 |
|  | (3.17) | (4.37) | (2.25) | (3.19) | (3.78) |
| Constant | 3.48 | 38.0*** | 66.7*** | 85.6*** | 95.0*** |
|  | (7.77) | (9.95) | (7.33) | (7.88) | (8.01) |

*Note*: Robust standard errors are in parentheses (* $p < 0.1$, ** $p < 0.05$, *** $p < 0.01$). We also control for parent's income, education level, nationality, number of siblings, relationship status, relationship with parents, number of close friends and school performance. These coefficients are not reported in the table but are available upon request.

symptoms. This suggests the presence of reporting heterogeneity in self-reported sleep difficulties in these two subgroups of individuals. College students with moderate depressive symptoms also tend to evaluate vignette 1 higher than college students without any depressive symptoms (6 points, $p < 0.05$).

Corresponding results when using PHQ-9 score instead of depressive symptoms as measure of mental health are presented in S2 Table. We show in the Appendix that PHQ-9 score has a statistically significant association with the evaluations of vignette 2 (0.55 points, $p < 0.01$) but has no significant statistical correlation (at conventional level) with the evaluations of vignettes 1, 3, 4 and 5. These results are very much in line with results presented in Table 3.

To further determine the existence of reporting heterogeneity, we add vignette evaluations directly in the sleep difficulties regression as control variables. Indeed, assuming that individuals use the same reporting scale to evaluate their sleep difficulties as the scale they use to evaluate the vignettes, reporting heterogeneity can be directly controlled for by adding vignette evaluations as regressors in the econometric specification. Table 4 reports the results of these

**Table 4. Linear regressions of sleep difficulties on our set of control variables, including the vignette evaluations in columns 2 and 4.**

|  | (1) Sleep difficulty | (2) Sleep difficulty | (3) Sleep difficulty | (4) Sleep difficulty |
|---|---|---|---|---|
| Female | 1.24 | 1.08 | 0.21 | 0.053 |
|  | (1.20) | (1.19) | (1.09) | (1.08) |
| Age | 0.20 | 0.18 | 0.14 | 0.12 |
|  | (0.39) | (0.39) | (0.36) | (0.36) |
| Mild depressive symptoms | 19.6*** | 19.6*** | 14.1*** | 14.1*** |
|  | (1.69) | (1.70) | (1.57) | (1.57) |
| Moderate depressive symptoms | 32.4*** | 30.9*** | 24.7*** | 23.2*** |
|  | (2.40) | (2.40) | (2.27) | (2.21) |
| Severe depressive symptoms | 37.4*** | 36.8*** | 28.5*** | 27.9*** |
|  | (3.69) | (3.64) | (3.40) | (3.28) |
| Vignette 1 |  | 0.098*** |  | 0.10*** |
|  |  | (0.035) |  | (0.032) |
| Vignette 2 |  | 0.091*** |  | 0.088*** |
|  |  | (0.026) |  | (0.024) |
| Vignette 3 |  | 0.032 |  | 0.047 |
|  |  | (0.040) |  | (0.035) |
| Vignette 4 |  | 0.075* |  | 0.072* |
|  |  | (0.044) |  | (0.039) |
| Vignette 5 |  | -0.0022 |  | 0.00078 |
|  |  | (0.044) |  | (0.041) |
| Take more than 20 mins to fall asleep |  |  | 17.7*** | 17.7*** |
|  |  |  | (1.17) | (1.15) |
| Sleep less than 7 hours |  |  | 6.76*** | 6.79*** |
|  |  |  | (1.42) | (1.41) |
| Low sleep efficiency |  |  | 9.07*** | 9.15*** |
|  |  |  | (1.48) | (1.48) |
| Constant | 17.5* | 5.38 | 11.9 | -1.21 |
|  | (9.93) | (10.4) | (9.18) | (9.68) |

*Note*: Robust standard errors are in parentheses (* $p < 0.1$, ** $p < 0.05$, *** $p < 0.01$). We also control for parent's income, education level, nationality, number of siblings, relationship status, relationship with parents, number of close friends and school performance. These coefficients are not reported in the table but are available upon request.

**Table 5. Linear regressions of sleep difficulties on our set of control variables, accounting for reporting heterogeneity.**

| | (1) "True" effect $b_0$ | (2) Reporting Heterogeneity $\gamma$ | (3) "True" effect $b_0$ | (4) Reporting Heterogeneity $\gamma$ |
|---|---|---|---|---|
| Female | 1.48 | -0.24 | 0.42 | -0.22 |
| | (1.25) | (0.56) | (1.14) | (0.56) |
| Age | 0.25 | -0.049 | 0.19 | -0.056 |
| | (0.41) | (0.18) | (0.38) | (0.18) |
| Mild depressive symptoms | 20.1*** | -0.51 | 14.5*** | -0.44 |
| | (1.78) | (0.72) | (1.66) | (0.75) |
| Moderate depressive symptoms | 29.8*** | 2.62** | 22.0*** | 2.70** |
| | (2.60) | (1.32) | (2.43) | (1.35) |
| Severe depressive symptoms | 35.3*** | 2.07 | 26.4*** | 2.12 |
| | (3.85) | (2.20) | (3.56) | (2.21) |
| Take more than 20 mins to fall asleep | | | 18.2*** | -0.52 |
| | | | (1.21) | (0.56) |
| Sleep less than 7 hours | | | 6.46*** | 0.29 |
| | | | (1.49) | (0.71) |
| Low sleep efficiency | | | 9.05*** | 0.015 |
| | | | (1.61) | (0.76) |
| Constant | 17.5* | | 11.9 | |
| | (9.88) | | (9.13) | |

*Note*: Cluster robust standard errors at the respondent level reported in parentheses (* $p < 0.1$, ** $p < 0.05$, *** $p < 0.01$). We also control for parent's income, education level, nationality, number of siblings, relationship status, relationship with parents, number of close friends, and school performance. These coefficients are not reported in the table but are available upon request. Columns 1 and 3 show the effects of the control variables on sleep difficulties, net of reporting heterogeneity, while columns 2 and 4 show the effects of the control variables on reporting heterogeneity.

regressions. Column 1 is the result of regressing sleep difficulties on our set of control variables (same regression as column 1 of Table 2) and column 2 represents the results of the same regression including this time vignettes evaluations as control variables. Although vignettes 1's and 2's evaluations are themselves highly significant (0.10 points with p<0.01 and 0.09 points with p<0.01, respectively), the inclusion of vignettes as control variables does not affect the associations between depressive symptoms and sleep difficulties. The conclusion is identical when we add vignette evaluations (column 4) to the model that also controls for sleep patterns and efficiency (column 3). Our results are robust to using PHQ-9 score as a continuous measure instead of descriptive depression categories, as shown in S2 Table. These results indicate that the associations between sleep difficulties and depressive symptoms remain large and statistically significant even after controlling for reporting heterogeneity.

Because there nonetheless appears to be some reporting heterogeneity in self-assessed sleep difficulties, as evidenced by the statistically significant effect of some of our vignettes, it would be interesting to understand what factors drive this reporting heterogeneity. Since we observe both self-assessed sleep difficulties and vignette evaluations, we can run a double-index model that allows us to estimate the associations between our control variables and both sleep difficulties (free from any reporting heterogeneity) and reporting heterogeneity itself, hence enabling us to determine the factors explaining reporting heterogeneity.

Results from this model are summarized in Table 5. Odd-numbered columns represent the "true" associations between our control variables, net of any reporting heterogeneity (vector of coefficients $b_0$), and sleep difficulties, and even-numbered columns show the associations

between our set of regressors and reporting heterogeneity (vector of coefficients $\gamma$). From column 1, it is clear that the associations between being mildly, moderately, and severely depressed and sleep difficulties remain very strong (with increases of 20.1, 29.8, and 35.3 points, respectively, compared to those with no depressive symptoms, all with p<0.01) and are very close to the ones obtained in column 2 of Table 4. The same is true when we include sleep patterns and efficiency in the model (column 3). Columns 2 and 4 show that individuals suffering from mild and severe depressive symptoms do not appear to evaluate the VAS scale differently from those with no depressive symptoms. Respondents with moderate depressive symptoms, however, do perceive the VAS scale differently, with their sleep difficulty evaluations being 2.6-2.7 points higher than what it "really should be". That being said, this effect is very small, less than 10% in magnitude, and the associations between sleep difficulties and depressive symptoms remain very strong even after controlling for reporting heterogeneity. Again, these results are robust to using PHQ-9 score as a continuous measure instead of descriptive depression categories (S2 Table).

## Discussion and conclusion

Many studies have analyzed the association between sleep difficulties and depression [2, 67, 68]. All found that individuals who suffer from depression have more severe sleep difficulties. However, very few studies have investigated whether that association could be explained, at least partially, by reporting heterogeneity [44]. Based on a sample of college students (18-25 years of age) asked to complete a survey online, we confirm that sleep difficulties and depressive symptoms are strongly related. Individuals with mild, moderate, and severe depressive symptoms rated their sleep difficulties to be 20, 32, and 37 points higher, on a 0-100 Visual Analogue Scale, compared with those who have no or minimal depressive symptoms. These associations hold even after accounting for differences in sleep patterns and efficiency.

We then ask whether this association finds its explanation, at least partially, in reporting heterogeneity. Our analysis does indicate the presence of reporting heterogeneity in self-assessed sleep difficulties. We show that college students who suffer from moderate depressive symptoms tend to be pessimistic about their sleep difficulties, rating them 2.6-2.7 points higher than if they were to use the same reporting scale as college students with no or minimal symptoms (reference group). Given the large magnitude of the associations between sleep difficulties and depressive symptoms, the difference in reporting patterns between college students with no or minimal symptoms and those with moderate depressive symptoms appear to be minor (less than 10% in magnitude). Interestingly, we do not find corresponding differences in reporting patterns between individuals with no or minimal depressive symptoms and those with severe depressive symptoms. Individuals with severe depressive symptoms appear to rate their sleep difficulties about 2 points higher than if they were to use the same reporting scale as those with no or minimal symptoms, but this effect is not statistically significant. A possible explanation for this absence of statistically significant reporting heterogeneity among individuals with severe depressive symptoms could stem from the fact that we have very few individuals in our sample that fall in that category (1.8% of our sample).

Our study investigates the existence of reporting heterogeneity in self-assessed sleep difficulties among college students with and without depressive symptoms. While focusing on this subpopulation is highly relevant because of the important changes in sleep patterns and dramatic increases in depression onset many college students are facing [91, 92]–with consequences for depression and bipolar disorder during adulthood [24]– our results only speak to the college student population and it is therefore difficult to generalize our findings to the overall population. This constitutes an important limitation and we hope to encourage further

researchers to explore whether our results hold in other settings and in the general population. Moreover, our results indicate minor differences in reporting behaviors between individuals with and without symptoms when it comes to evaluating their sleep difficulties. Future research could however explore the presence of reporting heterogeneity across these groups in other domains such as self-reported quality of life and well-being, for which corresponding reporting patterns are likely to be different.

Another limitation of our study is that our vignette adjustment to control for reporting heterogeneity rely on two assumptions, i.e., response consistency and vignette equivalence, that cannot be directly tested in our application. That being said, these two assumptions are very commonly used in the vignette literature [45, 47, 55, 62, 63] and we present in S3 and S4 Appendices. some evidence that suggests that they could hold in our setting as well. In addition, although widely used and validated, PHQ-9 is the sole instrument used in this study to determine the presence of depressive symptoms of our respondents and we therefore cannot rule out possible misclassification errors. Finally, we can not rule out the possibility that college students with depressive symptoms were less likely to participate in our study and complete our survey, which can lead to selection bias.

Overall, the results of our study suggest that comparisons of self-assessed sleep difficulties between college students are meaningful, even between those with and without depressive symptoms, and that reporting heterogeneity plays a minor role in explaining the relationship between sleep difficulties and depressive symptoms.

## Supporting information

**S1 Data.**
(ZIP)

**S1 Appendix. Vignettes used in this study.**
(PDF)

**S2 Appendix. The PHQ-9 questionnaire.**
(PDF)

**S3 Appendix. Response consistency.**
(PDF)

**S4 Appendix. Vignette equivalence.**
(PDF)

**S5 Appendix. Test of linear specification and fractional probit regression.**
(PDF)

**S1 Table. Descriptive statistics of sleep patterns.**
(PDF)

**S2 Table. PHQ-9 as a continuous measure.**
(PDF)

## Author Contributions

**Conceptualization:** Zhiyong Huang, Fabrice Kämpfen.

**Data curation:** Zhiyong Huang, Fabrice Kämpfen.

**Formal analysis:** Zhiyong Huang, Fabrice Kämpfen.

**Funding acquisition:** Zhiyong Huang, Fabrice Kämpfen.

**Investigation:** Zhiyong Huang, Fabrice Kämpfen.

**Methodology:** Zhiyong Huang, Fabrice Kämpfen.

**Project administration:** Zhiyong Huang, Fabrice Kämpfen.

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
