## [Decision Letter · Decision Letter 0]

28 Oct 2020

PONE-D-20-28953

The Association Between Depressive Symptoms and Self-reported Sleep Difficulties Among College Students: Truth or Reporting Bias?

PLOS ONE

Dear Dr. Kampfen,

Thank you for submitting your manuscript to PLOS ONE. After careful consideration, we feel that it has merit but does not fully meet PLOS ONE’s publication criteria as it currently stands. Therefore, we invite you to submit a revised version of the manuscript that addresses the points raised during the review process.

We look forward to receiving your revised manuscript.

Kind regards,

Jacob Resch, Ph.D.

Academic Editor

PLOS ONE

Journal Requirements:

Reviewers' comments:

Reviewer's Responses to Questions

**Comments to the Author**

1. Is the manuscript technically sound, and do the data support the conclusions?

Reviewer #1: Yes

Reviewer #2: Partly

2. Has the statistical analysis been performed appropriately and rigorously? 

Reviewer #1: Yes

Reviewer #2: Yes

3. Have the authors made all data underlying the findings in their manuscript fully available?

Reviewer #1: Yes

Reviewer #2: Yes

4. Is the manuscript presented in an intelligible fashion and written in standard English?

Reviewer #1: Yes

Reviewer #2: Yes

5. Review Comments to the Author

Reviewer #1: This study seeks to investigate whether the negative bias of depression could explain the known link with sleep difficulties. Swedish college students' subjectivity in assessing sleep did not influence the strong link between the degree of self-reported sleep difficulties and depressive symptoms. As was hypothesized, individuals with clinically significant depressive symptomatology displayed response heterogeneity on some vignettes. However, this slight variability was unable to fully account for the increased sleep difficulties reported in the more severely depressed students.

Strengths include the large sample size and robust vignette analyses. While the manuscript has a generally logical flow, it carries an inefficient writing style. Prose is wandering throughout and flawed at times. Use of a non-clinical sample is not consistently conveyed when framing the study’s purpose. The finding of response heterogeneity in the depressed group is an interesting one, which supports the negative bias hypothesis. Yet, this was not fully explored in the discussion. I also wonder if mediator/moderator relationships are possible given the vignette findings--an area for potential consideration and exploration.

Abstract

-“reliable” does not fit in the context of a significant association between vignette responses and depressive symptomatology

Data Availability

-Where have data been made available, as indicated in your statement? The instructions request completed sentences here.

Introduction

-Line 13: Omitting the phrase, “making the college student population vulnerable to mental health problems,” would more succinctly flow with the supportive points that follow. As is, it feels like the cart is before the horse.

-Lines 19-21: Numbers for statistics and for citations are confusing due to dashes here, which seem misformatted. This is also part of a lengthy sentence that could better connect the ideas in this paragraph and next with smaller phrases.

-Lines 39-43: To indicate discrepancy among these articles warrants, even if very brief, mention of possible or hypothesized explanation for these different findings.

-The introduction outlines the risk of mental health in adolescence but revision to the final paragraphs is first needed to clearly link this idea to the present non-clinical sample. Importantly, lines 91-92 and 115 should be corrected to indicate that participants consisted of both college students with and without elevated depressive symptoms.

-Line 112: Both a dash and comma are not needed here, and in fact omitting “-a continuous measure-,” would be more succinct.

Methods

-Sparing use of footnotes is preferable, as I do not see a clear need for using footnotes in the sections before and after but most notably in the methods (lines 121, 126, 133, 218). This information would be more efficient for readers if integrated within the body of the text.

-Lines 131-132: The phrase “that generate major changes both in terms of mental health and sleep patterns/circadian rhythm” warrants an addition citation. Consider rephrasing in a way that indicates a risk factor rather than a definitive causal link if no firm citation is provided/available.

-Lines 137 & 139: should be “asked”

-Line 206: “objective measures of sleep patterns” may lead readers to believe polysomnography or actigraphy is in use rather than vignettes.

Results

-Given report here that 25% of the sample had scores on the depressive screener reaching clinically significant levels, framing this sample as “depressed” here and elsewhere in the paper would seem confusing.

-Were the assumptions of linear regression analyses met?

-Line 309: should be “the time it takes individuals”

-Lines 323-236 and lines 400-102: Stating “do not appear to explain the relationship between depressive symptoms and sleep difficulties” and “the associations between sleep difficulties and depressive symptoms are unlikely to be severely biased” does not convey regression results truthfully. More accurate is the phrasing in lines 232-335, highlighting that there remains to be a significant association between sleep difficulty and depression groups even after accounting for these other variables.

-Lines 327 and 426-427: Rather than “mental health” the emphasis should be placed on your analyzing the PHQ-9 as a continuous measure vs. descriptive depression categories, and that findings were consistent.

-Line 407: “allows to” should be “allows us to”

Discussion

-Lacking is implications of results from the subset of the sample that carries severe depressive symptomatology, which is the group most prone to negative bias in reporting.

-Line 433: Change to “college students asked to complete a survey...”

-Lines 437-439: To be more succinct and avoid pitfalls described above, simplify this sentence to reflect that depression groups had greater sleep difficulty even after accounting for differences in sleep patterns and efficiency.

-Lines 441-444: omit “what it is when using the same reporting scale as” from this sentence, as this is implied.

-Lines 453-455 are the only reference to future research. What other areas for future study are implicated with these findings?

-Line 463: Again, “reliable” would be inaccurate given the significant response heterogeneity. Also change “suffering from” to “with and without” or similar phrase.

References:

-Calling out citations mid-sentence with a bracketed number (e.g., lines 39, 40, 42, 108, 140, 151, footnote 10) is less appealing than preceding this by an author name or a noun describing the type of work. I would not be surprised if this was inadvertently done by a citation manager.

Tables/Figures:

-Figures are missing y-axis labels.

-The meaning of “vignette restricted” in the supplemental figures is not clearly described.

-Avoid using a form of the word “control” twice in titles (e.g., Table 5).

Reviewer #2: Introduction:

General comments: The introduction is difficult to follow in that the author jumps around from mental health and sleep. While there are several variables (as examined in the methods/results) that may lead to depressive symptoms (such as family income and school performance), the PHQ9 does not assess any of those as they relate to depression. Furthermore, there is not sufficient discussion of the PHQ9 in the introduction to validate why it is the sole questionnaire being used to assess mental health in this study.

Line 8: Give age range for “young adults”

Line 11: Switching between “adolescence” and “young adults/college students” is confusing. I would decide on one term and be consistent with it. Also would advise defining what the age range is for which every population term you use

Lines 36-43: What are the objective measures you are referring to when comparing them with subjective measures?

Line 117: There is no hypothesis/outcome measures stated for this study

Methods: Of the 1813 respondents in your subject pool, did you only include those that recorded depressive symptoms on the PHQ9? Based on your description it is not clear.

How was the Qualtrics survey worded when it was distributed? Without knowing this, it may lead a reader to wonder if there is non-respondent bias.

Overall comments:

I appreciate the idea behind this study but feel it is not framed in an appropriate way. It may be better written from a perspective of the self-reported sleep habit of individuals with and without depressive symptoms. As it reads now, it is unclear if the authors compared those with depressive symptoms to those without, which I feel is an important aspect to establish. Second, as mentioned in the Methods comments, there is no clear description of how the recruitment via Qualtrics was done. It should be made clear if this survey was sent out to students informing them it was for a study regarding depression and/or mental health as it relates to sleep. If this was in fact how it was stated, there is a risk of non-respondent bias in that students who experienced depression/mental health issue may have been more likely to respond to the survey than those that did not experience these symptoms. If the overall goal is to see if the reporting behaviors are different amongst individual with depressive symptoms then there should be 2 groups being compared (those with and those without).

6. PLOS authors have the option to publish the peer review history of their article (what does this mean?). If published, this will include your full peer review and any attached files.

Reviewer #1: **Yes: **Kristin Wilmoth, PhD

Reviewer #2: No

---

## [Author Response · Author response to Decision Letter 0]

6 Dec 2020

Response letter to the reviewers’ comments on manuscript submission PONE-D-20-28953 entitled “The Association Between Depressive Symptoms and Self-reported Sleep Difficulties Among College Students: Truth or Reporting Bias?”

Dear Editor,

dear Reviewers:

Thank you very much for your valuable feedback on our manuscript PONE-D-20-28953. We appreciate the opportunity to revise and resubmit our manuscript to PLOS ONE. The provided comments raised some good points and were helpful to improve the manuscript. We hope that the revised paper will be acceptable for publication in PLOS ONE.

This letter outlines our changes in the manuscript in response to the reviewers’ comments and suggestions and provides specific answers to all issues raised in their reviews. For convenience, we first reproduce the reviewers’ comments and then provide corresponding answers after each comment in italics.

Reviewers' comments:

Reviewer #1: 

Overall assessment: Strengths include the large sample size and robust vignette analyses. While the manuscript has a generally logical flow, it carries an inefficient writing style. Prose is wandering throughout and flawed at times. Use of a non-clinical sample is not consistently conveyed when framing the study’s purpose. The finding of response heterogeneity in the depressed group is an interesting one, which supports the negative bias hypothesis. Yet, this was not fully explored in the discussion. I also wonder if mediator/moderator relationships are possible given the vignette findings--an area for potential consideration and exploration.

Authors’ response: 

Thank you very much for your very constructive comments. We revised our manuscript along the lines you suggest,

and we believe our paper is now much stronger than it was before. Thanks to your comments, the writing style is

more efficient. Importantly, we clarified that our study sample does not only include college students with

depressive symptoms but that students with no or minimal symptoms are also included in the analysis. In fact,

they are the reference group in all our analysis. We also expanded the discussion section, which now contains a

more in-depth discussion about the implications of the presence of reporting heterogeneity in sleep

difficulties among individuals with depressive symptom, albeit rather small. 

Abstract

1) “reliable” does not fit in the context of a significant association between vignette responses and depressive 

symptomatology

Authors’ response: 

Thank you for this comment. We meant to say that “unadjusted” comparisons of self-reported sleep difficulties

among college students can be trusted because our analysis does not detect the presence of large reporting

heterogeneity between individuals with and without depressive symptoms. That’s the reason why we wrote

in the previous version of the abstract that comparisons are “reliable”. We do understand that it might not be

entirely appropriate, however. We have therefore changed the last sentence of the abstract, which now reads

as follows: 

“This suggests that unadjusted comparisons of self-reported sleep difficulties between college students are

meaningful, even among individuals with depressive symptoms.”

Data Availability

2) Where have data been made available, as indicated in your statement? The instructions request completed

sentences here.

Authors’ response: 

We have included the data and the codes as Supporting Information files in our resubmission. 

Introduction

3) Line 13: Omitting the phrase, “making the college student population vulnerable to mental health problems,”

would more succinctly flow with the supportive points that follow. As is, it feels like the cart is before the horse.

Author’s response: 

Thank you for your suggestion. We removed the phrase following your suggestion. 

4) Lines 19-21: Numbers for statistics and for citations are confusing due to dashes here, which seem misformatted.

This is also part of a lengthy sentence that could better connect the ideas in this paragraph and next with smaller

phrases.

Author’s response: 

You are absolutely right. We have in fact remove the sentence “75% of individuals who are diagnosed with a

mental health disorder throughout their life have had their first onset by the age of 25”, which didn’t add much

substance to the argument and made the sentence very long. 

5) Lines 39-43: To indicate discrepancy among these articles warrants, even if very brief, mention of possible or.

hypothesized explanation for these different findings.

Author’s response: 

We added a couple of sentences about possible or hypothesized explanations for these findings whenever it is

possible. Specifically, we state that:

“It has been shown that there exist significant differences between objective and subjective measures of sleep 

difficulties [26, 36-40]. Buysse et al. (2008) have attributed the weak association between subjective and objective 

measures of sleep to whether one assesses habitual patterns of sleep versus sleep on a discrete occasion. Baker et al. 

(1999) have argued that the weak correlation between objective and subjective measures are due to between-

individual variation while Jean-Louis and Kripke (2000) have speculated that the subjective measures may be 

biased by the depressive mood of respondents.”

6) The introduction outlines the risk of mental health in adolescence but revision to the final paragraphs is first

needed to clearly link this idea to the present non-clinical sample. Importantly, lines 91-92 and 115 should be 

corrected to indicate that participants consisted of both college students with and without elevated depressive 

symptoms.

Authors’ response: 

You are right: lines 91-92 and 115 suggested that our study was exclusively based on a sample of

college students who suffer from depressive symptoms, whereas we do in fact compare the associations between 

depression and sleep patterns for students with and without depressive symptoms (students without depressive 

symptoms are the reference category in our analysis). This is a very important clarification to make and we thank 

you for pointing that out. 

We revised lines 91—92 as follows: “The objective of this study is to determine the existence of reporting 

heterogeneity in subjective assessment of sleep difficulties among college students, and explore whether this 

reporting heterogeneity is explained by differences in depressive symptoms.”

In the same vein, line 115 now reads as: “In addition, our study is the first to investigate the existence of 

reporting heterogeneity in subjective assessment of sleep difficulties among college students, which, as 

discussed above, is a strata of the population that is particularly prone to both sleep difficulties and mental 

disorders.”

Hopefully these changes have made the link between the first and last part of the introduction clearer. 

7) Line 112: Both a dash and comma are not needed here, and in fact omitting “-a continuous measure-,” would 

be more succinct.

Authors’ response: 

We made the change, thank you.

Methods

8) Sparing use of footnotes is preferable, as I do not see a clear need for using footnotes in the sections before and

after but most notably in the methods (lines 121, 126, 133, 218). This information would be more efficient for 

readers if integrated within the body of the text.

Authors’ response: 

Thank you for your comment. You are correct. In fact, PLOS ONE does not allow the use of footnotes. We have

therefore adapted our manuscript accordingly.

9) Lines 131-132: The phrase “that generate major changes both in terms of mental health and sleep

patterns/circadian rhythm” warrants an addition citation. Consider rephrasing in a way that indicates a risk factor 

rather than a definitive causal link if no firm citation is provided/available.

Authors’ response: 

We added a few citations that support our statement and replaced the word “generate” with “coincide with”

to avoid definitive causal link.

10) Lines 137 & 139: should be “asked”

Authors’ response: 

We made the change, thank you.

11) Line 206: “objective measures of sleep patterns” may lead readers to believe polysomnography or

actigraphy is in use rather than vignettes.

Authors’ response: 

We meant “objective measures of sleep patterns” as measures that are derived using objective scales (such as hours

of sleep) and hence not subject to reporting heterogeneity. But we understand that readers might interpret 

“objective measures of sleep patterns” as measures of sleep pattern that are not self-reported. We therefore

changed our sentence to make our point clearer. The corresponding sentence now reads as follows:

“We further investigate the relationship between sleep difficulties and depressive symptoms using multivariate 

linear regressions, including various socio-demographic characteristics and self-reported measures of sleep 

patterns that are not subject to reporting heterogeneity, e.g. hours of sleep as control variables.”

Results

12) Given report here that 25% of the sample had scores on the depressive screener reaching clinically significant 

levels, framing this sample as “depressed” here and elsewhere in the paper would seem confusing.

Authors’ response: 

We changed the introduction along the lines you suggested so that hopefully it is clear for the readers that our

sample include both students with and without depressive symptoms. The confusion was mainly coming from the

statement “among college students with depressive symptoms” which we now changed to “among college students”

in several instances in the paper.

13) Were the assumptions of linear regression analyses met?

Authors’ response: 

We used linear models because it is simple and easy to implement/interpret. However, as you pointed out, it requires

a couple of assumptions. In the revision, to safeguard our assumption of linear regression models, we implemented

a number of tests and robust estimations, for which the corresponding results are presented in S5 Appendix.

Specifically, we first use individual-clustered variance estimation to account for heteroskedasticity to take into

account that we have several observations for each respondents (self-report and vignette evaluations). Second, we

use regression specification-error test (RESET) to examine the linear assumption in the specification in which we

regress sleep difficulty on our set of control variables (econometric model displayed in Table 2 column 1). The

result of this test suggests no violation of the linear assumption (F(3, 1783) =1.24 ,p=0.29)). Finally, given that our

VAS measures are bounded between 0 and 100, we use fractional probit response estimator to estimate the

regression of sleep difficulty on control variables and compare the results with the corresponding linear

specification estimates. As shown in the table of S5 Appendix, the fractional probit model and linear model produce

very similar estimates.

We added a sentence in the main text that refers to these results in S5 Appendix.

14) Line 309: should be “the time it takes individuals”

Authors’ response: 

We made the change, thank you.

15) Lines 323-236 and lines 400-102: Stating “do not appear to explain the relationship between depressive 

symptoms and sleep difficulties” and “the associations between sleep difficulties and depressive symptoms 

are unlikely to be severely biased” does not convey regression results truthfully. More accurate is the phrasing in 

lines 232-335, highlighting that there remains to be a significant association between sleep difficulty and depression 

groups even after accounting for these other variables.

Authors’ response: 

What we meant by “do not appear to explain the relationship between depressive symptoms and sleep 

difficulties” is that including the various measures of sleeping patterns and efficiency does not seem to 

mediate/attenuate the associations between depressive symptom characteristics and self-assessed sleep 

difficulties. We however understand the lack of clarity and have therefore modified the sentence accordingly, 

which now reads as:

“Results show that including these three indicator variables in our statistical specification, although they are 

individually all positive and highly significant (p<0.01), do not appear to attenuate the associations between 

depressive symptoms and sleep difficulties.”

Regarding the second sentence you mention, we changed it to: “These results indicate that the associations 

between sleep difficulties and depressive symptoms remain large and statistically significant even after controlling

for reporting heterogeneity.”

We hope these statements convey regression results more truthfully.

16) Lines 327 and 426-427: Rather than “mental health” the emphasis should be placed on your analyzing the 

PHQ-9 as a continuous measure vs. descriptive depression categories, and that findings were consistent.

Authors’ response: 

You are correct the sentences in lines 327 and 426-427 are confusing and we therefore changed them along the 

lines you suggest. More specifically, the sentence that was in line 327 now reads as: 

“Note that our results are robust to using PHQ-9 score as a continuous measure instead of descriptive 

depression categories.”

The sentence that was in line 426-427 was replaced with:

 “Again, these results are robust to using PHQ-9 score as a continuous measure instead of descriptive 

depression categories (S2 Tables).”

17) Line 407: “allows to” should be “allows us to”

Authors’ response: We made the change, thank you.

Discussion

18) Lacking is implications of results from the subset of the sample that carries severe depressive symptomatology,

which is the group most prone to negative bias in reporting.

Authors’ response: 

Thank you for this comment. Our results indicate that sleep difficulties of college students with moderate

depressive symptoms would be 2.6-2.7 points lower if they were to use the “same” reporting scale as college

students with no or minimal symptoms (reference group). As indicated in our discussion section, given the 

magnitude of the association between depressive symptoms and sleep difficulties, the magnitude of the difference 

in reporting patterns between individuals with moderate symptoms and those with no or minimal symptoms 

appear to be rather negligible. That being said, our analysis does not detect any differences in reporting 

patterns between individuals with no symptoms and those with severe depressive symptoms. So it does not seem 

that individuals with severe depressive symptoms are the most prone to negative bias in reporting. We 

modified the paragraph in the discussion section to make this point clearer. 

A possible explanation for the absence of reporting heterogeneity among individuals with severe depressive 

symptoms could stem from the fact that we have very few individuals in our sample that fall in that category 

(1.8% of our sample). We mention that explanation in the discussion section as well. 

Here is the paragraph that we updated in the discussion section:

“We show that college students who suffer from moderate depressive symptoms tend to be pessimistic about 

their sleep difficulties, rating them 2.6-2.7 points higher than if they were to use the “same” reporting scale 

as college students with no or minimal symptoms (reference group). Given the large magnitude of the associations 

between sleep difficulties and depressive symptoms, the difference in reporting patterns between college 

students with no or minimal symptoms and those with moderate depressive symptoms appear to 0 be minor (less 

than 10% in magnitude). Interestingly, we do not find corresponding differences in reporting patterns 

between individuals with no or minimal depressive symptoms and those with severe depressive symptoms. 

Individuals with severe depressive symptoms appear to rate their sleep difficulties about 2 points higher than 

if they were to use the same reporting scale as those with no or minimal symptoms, but this effect is not statistically 

significant. A possible explanation for this absence of reporting heterogeneity among individuals with severe 

depressive symptoms could stem from the fact that we have very few individuals in our sample that fall in that 

category (1.8% of our sample).”

19) Line 433: Change to “college students asked to complete a survey...”

Authors’ response: 

We made the change, thank you.

20) Lines 437-439: To be more succinct and avoid pitfalls described above, simplify this sentence to reflect that 

depression groups had greater sleep difficulty even after accounting for differences in sleep patterns and efficiency.

Authors’ response: 

We change the text accordingly following your suggestion:

“Individuals with mild, moderate, and severe depressive symptoms rated their sleep difficulties to be 20, 32, and 37 

points higher, on a 0-100 Visual Analogue Scale, compared with those who have no or minimal depressive 

symptoms. These associations hold even after accounting for differences in sleep patterns and efficiency.”

21) Lines 441-444: omit “what it is when using the same reporting scale as” from this sentence, as this is implied.

Authors’ response: 

We made the change, thank you.

22) Lines 453-455 are the only reference to future research. What other areas for future study are implicated with

these findings?

Authors’ response: 

We expanded the discussion section by adding two sentences that discuss other research avenues that are 

implicated with our findings. The sentences we added are the following: 

“Moreover, our results indicate minor differences in reporting behaviors between individuals with and

without symptoms when it comes to evaluating their sleep difficulties. Future research could however explore the 

presence of reporting heterogeneity across these groups in other domains such as self-reported quality of life and 

wellbeing, for which corresponding reporting patterns are likely to be different.” 

23) Line 463: Again, “reliable” would be inaccurate given the significant response heterogeneity. Also change 

“suffering from” to “with and without” or similar phrase.

Authors’ response: 

We changed the last paragraph of our manuscript following your suggestions. It now reads as follows: 

Overall, the results of our study suggest that comparisons of self-assessed sleep difficulties between college students 

are meaningful, even between those with and without depressive symptoms, and that reporting heterogeneity plays a 

minor role in moderating the relationship between sleep difficulties and depressive symptoms. 

References:

24) Calling out citations mid-sentence with a bracketed number (e.g., lines 39, 40, 42, 108, 140, 151, footnote 10) 

is less appealing than preceding this by an author name or a noun describing the type of work. I would not be 

surprised if this was inadvertently done by a citation manager.

Authors’ response: 

We agree that calling out citations mid-sentence with bracket numbers instead of the classic “author (year)” 

format is suboptimal. We are using the PLOS ONE latex template which contains the “cite” package which 

apparently does not allow us to straightforwardly call out both “[number]” and “author [number]” citations at the same time. To address your comment, we have manually added the name of the author(s) and year of the study we are citing when citations appear in the middle of the sentence.

Tables/Figures:

25) Figures are missing y-axis labels.

Authors’ response: 

Sorry about this omission. The figures now contain the labels on the y-axis. 

26) The meaning of “vignette restricted” in the supplemental figures is not clearly described.

Authors’ response: 

Thank you for this comment. We added a short explanation in the text of the Appendix and in the notes of Figures 

S3A and S3B. The notes of the figures now include the following sentence: 

“The evaluations of the individuals who are included in that restricted sample are labeled “restricted'' in

the legend of the figure.” We hope that clarifies what we mean by “restricted”. 

27) Avoid using a form of the word “control” twice in titles (e.g., Table 5).

Authors’ response: 

We replaced “controlling” with “accounting” in Table 5 and in two tables in the Appendix.

Reviewer #2: 

1) The introduction is difficult to follow in that the author jumps around from mental health and sleep. While there are several variables (as examined in the methods/results) that may lead to depressive symptoms (such as family income and school performance), the PHQ9 does not assess any of those as they relate to depression. Furthermore, there is not sufficient discussion of the PHQ9 in the introduction to validate why it is the sole questionnaire being used to assess mental health in this study.

Authors’ response: 

We hope that the changes we made to the introduction thanks to reviewer 1 make it easier for readers to read the

introduction.

We opted for PHQ-9 to measure depressive symptoms of our respondents for several reasons. First, the PHQ-9 is a

well-validated instrument that is frequently used to screen for depression in primary care and non-clinical settings 

(Arrieta et al., 2017; Kohler, Payne, Bandawe, & Kohler, 2017; Spitzer, Williams, & Kroenke, 2014). Second, PHQ-

9 has been shown to have good validity and credibility as an instrument for depression among college students 

(Adewuya, Ola, & Afolabi, 2006; Huang, Kohler, & Kämpfen, 2020; Kim & Lee, 2019). 

Finally, the use of the computerized version of the PHQ-9 has been validated, which is essential for our survey that 

was conducted online (Erbe et al, 2016).

We added a sentence about PHQ-9 in the introduction, as well as more details in the “Measures on depressive

symptoms” subsection and acknowledged the limitations of the PHQ-9 instrument in the limitations. 

Given the time and financial constraints we were facing in online survey, opting the PHQ-9 instrument was a very 

natural choice to make. Nevertheless, we acknowledged the limitation of having used only one instrument to assess 

the depressive symptoms of our respondents in the discussion. We added:

“Though widely used and validated, PHQ-9 is the sole instrument used in this study to determine the depressive 

symptoms of our respondents and we therefore cannot rule out possible misclassification errors.”

References

Adewuya, A. O., Ola, B. A., & Afolabi, O. O. (2006). Validity of the patient health questionnaire (PHQ-9) as a screening tool for depression amongst Nigerian university students. Journal of Affective Disorders, 96(1–2), 89–93.

Arrieta, J., Aguerrebere, M., Raviola, G., Flores, H., Elliott, P., Espinosa, A., … others. (2017). Validity and utility of the Patient Health Questionnaire (PHQ)-2 and PHQ-9 for screening and diagnosis of depression in rural Chiapas, Mexico: A cross-sectional study. Journal of Clinical Psychology, 73(9), 1076–1090.

Erbe, D., Eichert, H. C., Rietz, C., & Ebert, D. (2016). Interformat reliability of the patient health questionnaire: validation of the computerized version of the PHQ-9. Internet Interventions, 5, 1-4.

Huang, Z., Kohler, I. V., & Kämpfen, F. (2020). A Single-Item Visual Analogue Scale (VAS) Measure for Assessing Depression Among College Students. Community Mental Health Journal, 56(2), 355–367. https://doi.org/10.1007/s10597-019-00469-7

Kim, Y. E., & Lee, B. (2019). The psychometric properties of the patient health questionnaire-9 in a sample of Korean university students. Psychiatry Investigation. https://doi.org/10.30773/pi.2019.0226

Kohler, I. V, Payne, C. F., Bandawe, C., & Kohler, H.-P. (2017). The demography of mental health among mature adults in a low-income, high-HIV-prevalence context. Demography, 54(4), 1529–1558.

Spitzer, R. L., Williams, J. B. W., & Kroenke, K. (2014). Test review: Patient Health Questionnaire--9 (PHQ-9). Rehabilitation Counseling Bulletin, 57(4), 246–248.

2) Line 8: Give age range for “young adults”

Authors’ response: 

We added the age range for “young adults”. 

3) Line 11: Switching between “adolescence” and “young adults/college students” is confusing. I would decide on one term and be consistent with it. Also would advise defining what the age range is for which every population term you use

Authors’ response: 

Thank you for this comment. We used “young adults/college students” consistently in the revised manuscript. We 

also added the age range when we referred to “young adults/college students” in the introduction, methods and 

discussions. 

4) Lines 36-43: What are the objective measures you are referring to when comparing them with subjective measures?

Authors’ response: 

Polysomnographic recordings of sleep and actigraphy are the two usual techniques that are used to objectively 

measure sleep quality. We now explicitly name these two techniques in the text, thank you.

5) Line 117: There is no hypothesis/outcome measures stated for this study

Authors’ response: 

We added our hypothesis explicitly in the introduction section. We now state the following:

“We hypothesize that college students with and without depressive symptoms adopt different reporting scales when 

assessing their sleep difficulties, to an extent that is hard to evaluate a priori.”

Methods: 

6) Of the 1813 respondents in your subject pool, did you only include those that recorded depressive symptoms on the PHQ9? Based on your description it is not clear. How was the Qualtrics survey worded when it was distributed? Without knowing this, it may lead a reader to wonder if there is non-respondent bias.

Authors’ response: 

We indeed included all students --with or without depressive symptoms-- in our study. As indicated in Table 1, 76% 

of our 1813 respondents display no depressive symptoms. Students with no or minimal depressive symptoms are the 

reference category in our analysis.

As pointed out by yourself and Reviewer1, the previous version of our manuscript did not make that clear enough 

and we have revised our manuscript accordingly. Please see Reviewer 1 points 6 and 12 above.

Moreover, we added in the method section information about how students were recruited to participate in our 

study. The sentence we added is the following: 

“The invitation emails we sent to students and the introductory page of our online survey did not mention the 

specific aim of the study but referred to it as a study on their quality of life in general.” 

Finally, we can’t rule out the possibility that college students with depressive symptoms were less likely to 

participate in our study and complete our survey. Selection bias can arise if, conditional on observables, 

participants and non-participants are different in terms of sleep difficulties or reporting patterns. We acknowledged 

this limitation in the discussion section. 

Overall comments:

I appreciate the idea behind this study but feel it is not framed in an appropriate way. It may be better written from a perspective of the self-reported sleep habit of individuals with and without depressive symptoms. As it reads now, it is unclear if the authors compared those with depressive symptoms to those without, which I feel is an important aspect to establish. Second, as mentioned in the Methods comments, there is no clear description of how the recruitment via Qualtrics was done. It should be made clear if this survey was sent out to students informing them it was for a study regarding depression and/or mental health as it relates to sleep. If this was in fact how it was stated, there is a risk of non-respondent bias in that students who experienced depression/mental health issue may have been more likely to respond to the survey than those that did not experience these symptoms. If the overall goal is to see if the reporting behaviors are different amongst individual with depressive symptoms then there should be 2 groups being compared (those with and those without).

Authors’ response:

Thanks a lot for your positive assessment of our study and for your constructive comments. We hope our manuscript

is now clearer and the information provided regarding our study sample (which include both individuals with and 

without depressive symptoms) and the recruitment procedure is more complete and better communicated. 

Journal’s comments:

 Authors’ response:NA

Authors’ response: 

We have uploaded the data we used in our study as well as the corresponding codes as Supporting information files.

Authors’ response:

We have included a caption for each figure These captions are in bold and are located right after the [INSERT 

FIGURE XX HERE] in the tex file. Thank you.

---

## [Decision Letter · Decision Letter 1]

14 Jan 2021

PONE-D-20-28953R1

The Association Between Depressive Symptoms and Self-reported Sleep Difficulties Among College Students: Truth or Reporting Bias?

PLOS ONE

Dear Dr. Kämpfen,

Thank you for submitting your manuscript to PLOS ONE. After careful consideration, we feel that it has merit but does not fully meet PLOS ONE’s publication criteria as it currently stands. Therefore, we invite you to submit a revised version of the manuscript that addresses the points raised during the review process.

We believe your article will be of great interest to our readership, however, more editing based on the comments below is required. When you reviewing your submission, please pay special attention to reviewer one's comments regarding being more succinct as well as being more specific when it comes to the description of your methods. 

We look forward to receiving your revised manuscript.

Kind regards,

Jacob Resch, Ph.D.

Academic Editor

PLOS ONE

Reviewers' comments:

Reviewer's Responses to Questions

**Comments to the Author**

1. If the authors have adequately addressed your comments raised in a previous round of review and you feel that this manuscript is now acceptable for publication, you may indicate that here to bypass the “Comments to the Author” section, enter your conflict of interest statement in the “Confidential to Editor” section, and submit your "Accept" recommendation.

Reviewer #1: (No Response)

2. Is the manuscript technically sound, and do the data support the conclusions?

Reviewer #1: Yes

3. Has the statistical analysis been performed appropriately and rigorously? 

Reviewer #1: Yes

4. Have the authors made all data underlying the findings in their manuscript fully available?

Reviewer #1: Yes

5. Is the manuscript presented in an intelligible fashion and written in standard English?

Reviewer #1: Yes

6. Review Comments to the Author

Reviewer #1: The authors have addressed those areas identified during my last review. In particular, I appreciate their improved explanation of the depression groups used in their models. I offer some further points of clarification, with line numbers corresponding to the version with tracked changes in blue. These minor suggestions within the methods and results are primarily for readability and flow to this paper, which I feel will be of interest to the readership of PLOS One. I thank the authors and editors of PLOS One for allowing me to participate in this review.

Methods

-Line 134: Consider removing "against payment" or clarifying. If there is a lottery, I assume that not all students will be guaranteed renumeration.

-Lines 159-162: This sentence could be shortened. For example, "This age restriction also corresponds to late adolescent/young adulthood [69], which represents..."

-Lines 163-164: Removing the sentence "This adolescent/young adulthood age period is highly coincidental with the age distribution of our sample of college students," could make for a more succinct paragraph without loss of information.

-Line 168: Change "sample consisted in" to "sample consisted of..."

-Line 215: There is an extra space; please correct to "points."

-Lines 230 & 231: Should be "As a robustness...as a measure..."

-Line 232: Clarify what is meant by "alternative specification." The authors could rephrase to, "We show that our results when analyzing the presence of depressive symptoms are similar when analyzing depressive symptom severity," if this is accurate.

-Lines 253-257: Could shorten this sentence to: "This assumes that respondents use the same reporting scale when evaluating their own sleep difficulties as the scale they use when evaluating the sleep difficulties of the person described in the anchoring vignettes."

-Line 258: Should be "does not allow us to..."

-Lines 259-270: I see an opportunity here for the authors to be more concise. For example: "Another way to disentangle reporting heterogeneity and sleep difficulties is to perform a double-index model [60]. This model exploits respondents' evaluations of sleep difficulties for themselves and for vignettes, assuming that individuals use the same reporting scales. One can control for reporting heterogeneity as one controls for individual fixed effects in panel data models (where several observations are recorded for the same individual over time). The advantage is that the model allows us to not only control for reporting heterogeneity but also estimate the characteristics of the individuals that drive reporting heterogeneity."

Results

-Lines 349 & 356: I recommend avoiding the word "association," which is vague, and "decreasing" to describe the coefficients, as it may be confusing if these coefficients reflect an increased risk of sleep difficulties. Consider rephrasing such as, "For instance, the increase in sleep difficulties for those suffering from severe (moderate) depressive symptoms, compared to those with no depressive symptoms, only changed from ### points (p<0.01) to ### (p<0.01)," if this is accurate.

-Line 361: should be "the time it takes individuals"

-Line 469: Again, further explanation of the coefficients would seem warranted. Perhaps change to: "(with increases of 20.1, 29.8, and 35.3 points, respectively, compared to those with no depressive symptoms, all with p<0.01)"

7. PLOS authors have the option to publish the peer review history of their article (what does this mean?). If published, this will include your full peer review and any attached files.

Reviewer #1: **Yes: **Kristin Wilmoth, PhD

---

## [Author Response · Author response to Decision Letter 1]

15 Jan 2021

Response letter to the reviewer’s comments on manuscript submission PONE-D-20-28953R1 entitled “The Association Between Depressive Symptoms and Self-reported Sleep Difficulties Among College Students: Truth or Reporting Bias?”

Dear Editor,

dear Reviewer:

Thank you again for your valuable feedback on our manuscript PONE-D-20-28953R1. We really appreciate the time Reviewer 1 has taken to provide us with her valuable feedback and comments that greatly improved the readability and flow of our manuscript. We hope that the revised paper will be acceptable for publication in PLOS ONE. This letter outlines our changes in the manuscript in response to Reviewer 1’s comments and suggestions and provides specific answers to all issues raised in her reviews. For convenience, we first reproduce Reviewer 1’s comments and then provide corresponding answers after each comment in italics.

Reviewer's comments:

Reviewer #1: 

Methods

1) -Line 134: Consider removing "against payment" or clarifying. If there is a lottery, I assume that not all students will be guaranteed renumeration.

Authors’ answer: We removed “against payment”. You are correct: not all students were remunerated for their participation.

2) -Lines 159-162: This sentence could be shortened. For example, "This age restriction also corresponds to late adolescent/young adulthood [69], which represents..."

Authors’ answer: Done, thank you!

3) -Lines 163-164: Removing the sentence "This adolescent/young adulthood age period is highly coincidental with the age distribution of our sample of college students," could make for a more succinct paragraph without loss of information.

Authors’ answer: You are correct. That sentence was superfluous and didn’t provide additional information. 

4) -Line 168: Change "sample consisted in" to "sample consisted of..."

Authors’ answer: Done!

5) -Line 215: There is an extra space; please correct to "points."

Authors’ answer: Done! 

6) -Lines 230 & 231: Should be "As a robustness...as a measure..."

Authors’ answer: Done! 

7) -Line 232: Clarify what is meant by "alternative specification." The authors could rephrase to, "We show that our results when analyzing the presence of depressive symptoms are similar when analyzing depressive symptom severity," if this is accurate.

Authors’ answer: Yes, this is accurate. We indeed meant that our results are robust to using either PhQ-9 as a continuous or as a categorical variable. We changed our sentence accordingly. 

8) -Lines 253-257: Could shorten this sentence to: "This assumes that respondents use the same reporting scale when evaluating their own sleep difficulties as the scale they use when evaluating the sleep difficulties of the person described in the anchoring vignettes."

Authors’ answer: The second part of the original sentence was indeed redundant, and we therefore modified the text along the lines you suggest. 

9) -Line 258: Should be "does not allow us to..."

Authors’ answer: Done! 

10) -Lines 259-270: I see an opportunity here for the authors to be more concise. For example: "Another way to disentangle reporting heterogeneity and sleep difficulties is to perform a double-index model [60]. This model exploits respondents' evaluations of sleep difficulties for themselves and for vignettes, assuming that individuals use the same reporting scales. One can control for reporting heterogeneity as one controls for individual fixed effects in panel data models (where several observations are recorded for the same individual over time). The advantage is that the model allows us to not only control for reporting heterogeneity but also estimate the characteristics of the individuals that drive reporting heterogeneity."

Authors’ answer: Thank you for this suggestion. We modified the paragraph following your suggestion.

Results

11) -Lines 349 & 356: I recommend avoiding the word "association," which is vague, and "decreasing" to describe the coefficients, as it may be confusing if these coefficients reflect an increased risk of sleep difficulties. Consider rephrasing such as, "For instance, the increase in sleep difficulties for those suffering from severe (moderate) depressive symptoms, compared to those with no depressive symptoms, only changed from ### points (p<0.01) to ### (p<0.01)," if this is accurate.

Authors’ answer: You are right: we should be clearer about the fact that the coefficients reflect changes relative to the reference group, that is individuals with no or minimal symptoms. We modified the two sentences on lines 349 and 359 following your suggestions. Thanks again.

12) -Line 361: should be "the time it takes individuals"

Authors’ answer: Done! 

13) -Line 469: Again, further explanation of the coefficients would seem warranted. Perhaps change to: "(with increases of 20.1, 29.8, and 35.3 points, respectively, compared to those with no depressive symptoms, all with p<0.01)"

Authors’ answer: You are right. We should have explicitly mentioned that the effects (coefficients) were relative to the reference group (individuals with no depressive symptoms). We followed your suggestions and change the corresponding sentence accordingly.

---

## [Editor Report · Decision Letter 2]

19 Jan 2021

The Association Between Depressive Symptoms and Self-reported Sleep Difficulties Among College Students: Truth or Reporting Bias?

PONE-D-20-28953R2

Dear Dr. Kämpfen,

We’re pleased to inform you that your manuscript has been judged scientifically suitable for publication and will be formally accepted for publication once it meets all outstanding technical requirements.

Kind regards,

Jacob Resch, Ph.D.

Academic Editor

PLOS ONE

---

## [Editor Report · Acceptance letter]

25 Jan 2021

PONE-D-20-28953R2 

The Association Between Depressive Symptoms and Self-reported Sleep Difficulties Among College Students: Truth or Reporting Bias? 

Dear Dr. Kämpfen:

I'm pleased to inform you that your manuscript has been deemed suitable for publication in PLOS ONE. Congratulations! Your manuscript is now with our production department. 

Kind regards, 

on behalf of

Dr. Jacob Resch 

Academic Editor

PLOS ONE